# Neuropathology of Central and Peripheral Nervous System Lymphoma in Dogs and Cats: A Study of 92 Cases and Review of the Literature

**DOI:** 10.3390/ani13050862

**Published:** 2023-02-27

**Authors:** Niccolò Fonti, Francesca Parisi, Çağla Aytaş, Sara Degl’Innocenti, Carlo Cantile

**Affiliations:** 1Department of Veterinary Sciences, University of Pisa, Viale delle Piagge n. 2, 56124 Pisa, Italy; 2School of Veterinary Medicine and Science, University of Nottingham, Loughborough LE12 5RD, UK

**Keywords:** cats, central nervous system lymphoma, dogs, phenotype, pathological pattern, peripheral nervous system lymphoma

## Abstract

**Simple Summary:**

Nervous system lymphoma (NSL) is reported to be uncommon in dogs and rare in cats. The literature on this topic is fragmentary with heterogeneous results. Current knowledge is based on a few case series and mostly on single case reports. Therefore, the aim of our study was to retrospectively analyze 92 cases of canine and feline NSL, collecting data on breed, age, gender, clinical signs, type, neurolocalization, and assessing pathological patterns and phenotype. Finally, our results were compared with previously published studies and an extensive review of the literature was provided.

**Abstract:**

The literature about nervous system lymphoma (NSL) in dogs and cats is fragmentary, based on a few case series and case reports with heterogeneous results. The aim of our study was to retrospectively analyze 45 cases of canine and 47 cases of feline NSL and compare our results with previously reported data, also providing an extensive literature review. Breed, age, gender, clinical signs, type, and neurolocalization were recorded for each case. The pathological patterns and phenotype were assessed by histopathology and immunohistochemistry. The occurrence of central and peripheral NSL was similar between the two species in both primary and secondary types. NSL occurred with a slightly higher prevalence in Labrador Retrievers, and spinal cord lymphoma (SCL) was associated with young age in cats. The most frequent locations were the forebrain in dogs and the thoracolumbar segment in cats. Primary central nervous system lymphoma (CNSL) in cats most frequently involved the forebrain meninges, particularly as a B-cell phenotype. Peripheral NSL mostly affected the sciatic nerve in dogs and had no preferred location in cats. Nine different pathological patterns were identified, with extradural as the most prevalent SCL pattern in both species. Finally, lymphomatosis cerebri was described for the first time in a dog.

## 1. Introduction

Primary central nervous system lymphoma (CNSL) is a relatively uncommon form of tumor in dogs and is rare in cats, accounting for approximately 4% of all canine lymphomas [1] and less than 3% of cats with primary CNS tumors [2]. Lymphoma within the CNS develops most commonly as part of a multicentric process in both species with a reported prevalence up to 12% [3] or even nearly 30% [4] in dogs, and 7.8% [2] to 14.4% [5] in cats.

CNSL is believed to be more common in adult dogs with a mean age of onset of 7.4 years (range 3–11). Although there is no breed predisposition, Rottweiler dogs seem to have a slightly higher risk of developing CNS lymphoma [3,6]. The mean age of cats with prevalent location in the spinal cord is reported to be 6.3 years (range 2–8) [7] and no breed prevalence is recognized, with domestic shorthair cats being the most represented [2,5,8].

Peripheral nervous system lymphoma (PNSL) is less commonly encountered than CNSL. In dogs, it is mostly secondary [9,10,11,12,13], whereas in cats the primary type is most frequently recorded [2,8,14,15,16,17,18] and is considered the most common secondary tumor involving the PNS [15,19]. However, the incidence of both CNS and PNS lymphoma may vary depending on the tumor location in the neuraxis and spinal and cranial nerves [20,21].

The phenotype in canine CNSL is almost equally represented between B-cell [4,20,22,23,24,25,26,27] and T-cell [4,20,24,26,27,28,29,30,31,32,33,34], and less than 4% are reported as non-B non-T cell. B-cell lymphoma, especially diffuse large B-cell lymphoma, predominates in secondary types [26]. In cats, B-cell lymphoma [2,8,18,21,24,35,36,37,38,39,40] is more frequently detected than T-cell lymphoma [2,8,17,18,21,41,42,43].

Studies reporting a consistent number of cases with detailed neurolocalization and lesion distribution, as well as pathological patterns and immunophenotyping, are limited, with most of the existing literature being represented by case reports with fragmented information and therefore not always comparable. Given the lack of consistency in the evaluation of nervous system lymphoma in previous studies, the aims of this retrospective study were to determine the specific location, pathological patterns, phenotype, and prevalence of primary versus secondary, central and peripheral nervous system lymphomas in dogs and cats, evaluating 92 cases and comparing the results with the available literature.

## 2. Materials and Methods

### 2.1. Caseload

Ninety-two cases of central and peripheral nervous system lymphoma diagnosed between 1994 and 2022 (45 in dogs and 47 in cats) were retrieved from the archive of the Neuropathology Laboratory of the Department of Veterinary Sciences of Pisa University. Signalments including breed, age, gender, main neurological signs, and tissue sample origin (biopsy or necropsy) are reported in Table 1. Specimens were obtained either from surgical biopsy or post-mortem examination and routinely formalin-fixed and paraffin wax-embedded and processed for histology. For each case, the type (primary or secondary/multicentric), the anatomical sites (intracranial, intraspinal, cranial, and spinal nerves), and the precise location and distribution were also recorded. In primary neoplasia cases, the extra-neural system involvement was assessed clinically or after complete post-mortem examination in case of death, as previously reported by other authors [2,8].

### 2.2. Histopathology and Immunohistochemistry

Four-µm tissue sections were stained with hematoxylin and eosin (H&E), Luxol fast blue, and immunohistochemical methods. The histological classification and grading of the tumors were assessed according to the World Health Organization criteria [44]. Immunoperoxidase was performed using a rabbit polyclonal antibody anti-human-CD20 (1:200, Thermoscientific, Fremont, CA, USA) and a rat monoclonal antibody anti-human-Paired Box 5 (PAX5, 1:250, Abcam, Cambridge, MA, USA) as B-cell markers, and a rabbit polyclonal antibody anti-human-CD3 (1:200, Dako, Glostrup, Denmark) as T-cell marker. Antigen retrieval for CD3 and PAX5 was performed by heat-induced epitope retrieval in citrate buffer at pH 6.0. No retrieval was performed for anti-CD20 antibody. Sections were pretreated with 1% H_2_O_2_ in PBS for 10 min to quench endogenous peroxidase activity, then rinsed with 0.05% Triton-X (TX)-100 in phosphate-buffered saline (PBS) (3 × 10 min) and blocked for 1 h with 5% normal horse serum (PK-7200, Vector Labs, Burlingame, CA, USA) in PBS. The sections were then incubated overnight at 4 °C in a solution of the antibodies with 2% normal horse serum and 0.05% TX-100 in PBS. Sections were then rinsed in PBS (3 × 10 min), incubated for 20 min with universal biotinylated anti-mouse/rabbit IgG (Vectastain Kit, PK-7200, Vector Labs), and then with ABC reagent (Vectastain Kit, PK-7200, Vector Labs). Sections were again rinsed in PBS (3 × 10 min). The immunoreactivity was detected by the streptavidin-biotin peroxidase method (Streptavidin Peroxidase, ThermoFisher Scientific, Fremont, CA, USA), using 3,3′-diaminobenzidine as chromogen. Negative controls were obtained by replacing the primary antibody with an irrelevant, isotype-matched antibody and with an anti-serum. Canine and feline normal lymph nodes were used as positive control.

### 2.3. Pathological Patterns

Based on the histopathological location and distribution, the tumors were divided into the following pathological patterns: intraparenchymal mass (IP) (or intramedullary when in the spinal cord; IM), extraparenchymal (comprising both meningeal masses and leptomeningeal lymphomatosis; MM and LL, respectively), intravascular (IVL), lymphomatosis cerebri (LC), extradural (ED), intradural-extramedullary (ID-EM), and involving cranial and/or spinal nerves. In this latter condition, the term neurolymphomatosis (NL) was adopted, based on the presence of an infiltrating monomorphic population of malignant lymphoid cells within the endoneurium and perineurium of peripheral nerves [10,11]. To compare our results with the literature, we grouped the affected nerves in four compartments: (1) cranial nerves; (2) brachial plexus and/or its branches; (3) spinal roots, spinal nerves, and spinal ganglia; and (4) sciatic plexus and/or its branches. Additional localization of neoplastic cells included choroid plexus, pituitary gland, and surrounding soft tissue of spinal cord and nerves.

## 3. Results

Dogs affected by lymphoma had a mean age of 6.8 ± 3.8 years (ranging from 10 months to 14 years). There were 24/45 (53.3%) males and 21/45 (46.7%) females. Ten dogs were mixed breed, while the remaining were pure breed, namely Labrador Retriever (n = 5), Boxer (n = 4), German Shepherd (n = 3), Cane Corso (n = 3), Border Collie (n = 2), Hound (n = 2), Rottweiler (n = 2), and one of each of the following breeds: Epagneul Breton, Golden Retriever, Shih-tzu, Bulldog, West Highland White Terrier, Yorkshire Terrier, Dachshund, Beagle, Staffordshire Terrier, Bergamasco Shepherd, English Cocker Spaniel, Amstaff, Great Dane, and Pointer.

Cats affected by lymphoma had a mean age of 8.3 ± 4.4 years (ranging from 3 months to 15 years). There were 22/47 males (46.8%) and 25/47 females (53.2%). Most of the cats were European Shorthair (n = 40), while the remaining seven cats belonged to the following breeds: Persian, Ragdoll, Maine Coon, Norwegian Forest Cat, Carthusian, Siamese, and European Longhair.

The type, anatomical site, location, distribution, pathological pattern, nuclear size, grading, and immunophenotype of the lymphomas affecting the 92 animals are summarized in Table 2.

In dogs, 22/45 (48.9%) lymphomas were intracranial, 14/45 (31.1%) were located within the spinal canal, and 9/45 (20%) were limited to the peripheral nervous system (PNS). In four cases, a simultaneous CNS and PNS involvement was observed. In cats, 24/47 (51%) lymphomas were located intracranially, 21/47 (44.7%) in the spinal canal, and 7/47 (14.9%) in the PNS. Within these cases, in three cats lymphoma was detected both in the brain and spinal cord, and in another two cats in the brain together with cranial and spinal nerves. A total of 24/45 (53.3%) canine lymphomas were classified as primary forms, and 21/45 (46.7%) had multicentric involvement. In cats, 27/47 (57.4%) and 20/47 (42.6%) lymphomas were classified as primary and multicentric forms, respectively. Regarding the phenotype, the majority (24/45; 53.3%) of canine lymphomas had a B-cell phenotype, while 15/45 (33.3%) a T-cell phenotype, and 6/45 (13.4%) were non-B non-T cell lymphoma. In feline lymphomas, 25/47 (53.2%), 18/47 (38.3%), and 4/47 (8.5%) had B-cell, T-cell, and non-B non-T cell phenotypes, respectively.

### 3.1. Intracranial Lymphoma

The 27% (6/22) of canine and 20.8% (5/24) of feline intracranial lymphomas consisted of an intraparenchymal neoformation primarily but not exclusively located in the forebrain in both species (Figure 1A), and mostly as part of multicentric lymphoma in both dogs (five out of six; 83.3%) and cats (three out of five; 60.0%).

Grossly, focal or multifocal, whitish-grey masses accompanied by swelling of the peritumoral parenchyma and occasionally hemorrhagic areas were observed. The histopathological pattern was characterized by dense sheets of perivascularly arranged malignant cells that invaded the surrounding neuroparenchyma, creating highly cellular areas (Figure 1B). In one cat (#46), the neoplastic infiltration involved both the optic nerves and extended into the diencephalon. B-cell, T-cell, and non-B non-T phenotypes were observed in both species. In dogs, four out of six (66.7%) cases were B-cell, one out of six (16.7%) was T-cell, and one out of six (16.7%) was non-B non-T cell lymphoma. In cats, three out of five (60.0%) were T-cell and two out of five (40.0%) were B-cell lymphomas.

In one dog (#39), there was a grayish discoloration of large areas of the telencephalic white matter associated with bilateral loss of distinction between gray and white matter (Figure 2A). Moreover, there was a blurring appearance of the hippocampal layers and cerebral sulci. No space-occupying lesions were detected in any areas of the brain. Histopathologically, the neoplastic cells diffusely infiltrated the white matter tracts and the perivascular spaces (Figure 2B) and showed large, round nuclei with a thin rim of slightly eosinophilic cytoplasm. Anisokaryosis was moderate with two to three mitoses for high power field (HPF, 400×, 2.37 mm^2^), and apoptotic figures were observed. Frequently, neurons were normal, even if extensively surrounded by neoplastic cells. Most malignant cells were positively immunolabelled with anti-CD3 marker and a small number of CD20-positive B-lymphocytes were observed within the perivascular cell infiltrate (Figure 2C,D). Within the most severely affected areas, there was activation and proliferation of microglial cells. The morphological features of this case were consistent with lymphomatosis cerebri (LC).

Almost half of the canine intracranial lymphomas (10/22; 45.4%) occurred as intravascular lymphoma (IVL), characterized by aggregation of neoplastic cells within the blood vessel lumen and scant invasion of the surrounding neuroparenchyma (Figure 3A). Multifocal thrombosis, hemorrhage, and necrosis of different regions of the brain were consistently associated with IVL. Detailed features of these 10 cases were reported in a previous study [45].

Intracranial lymphomas were characterized in many cases by prominent meningeal involvement, resulting in an extra-axial location (extra-axial lymphoma, EAL). Meningeal lymphoma was detected in 15/24 (58.3%) cats and in 2/22 (9.1%) dogs. These two canine tumors were multicentric, while 9/15 (60.0%) of feline cases were primary. This pathological pattern appeared as a well-defined, focal, irregular mass within the leptomeninges, resulting in a space-occupying lesion with compression of the underlying nervous tissue (Figure 3B). The cells exhibited mild to severe pleomorphism and typical lymphoid appearance. In most cases, despite well-defined margins, the meningeal masses were found in association with perivascular aggregates of neoplastic lymphoid cells within the neuroparenchyma, showing an intraparenchymal pattern. Both B-cell and T-cell phenotypes were observed in the two canine tumors. In cats, 9/15 of lymphomas had a B-cell phenotype (60.0%), whereas 4/15 (26.7%) and 2/15 (13.3%) showed a T-cell and non-B non-T phenotype, respectively. With regard to location, all feline tumors involved the forebrain, mainly the leptomeninges of the frontal lobes. Additionally, in two feline cases, the meningeal masses were associated with an intradural-extramedullary (#77) and an extradural (#79) spinal cord lymphoma.

Four cases of leptomeningeal lymphomatosis (LL) were observed; two were in dogs and two were in cats. In this pattern, neoplastic lymphocytes showed widespread diffusion predominantly within the subarachnoid space, infiltrating in some cases the subpial neuroparenchyma and perivascular spaces (Figure 4A). The canine forms were a multicentric T-cell lymphoma affecting the brain (#40) and a multicentric B-cell lymphoma restricted to the spinal cord (#35). The feline forms were a primary T-cell intracranial neoplasia (#88) and a multicentric B-cell lymphoma affecting both the brain and the spinal cord (#49, Figure 4B).

### 3.2. Intraspinal Lymphoma

Extradural (ED) and intramedullary (IM) pathological patterns of spinal cord lymphoma (SCL) were observed in both species. A total of 81% (17/21) of all feline SCL described in this study were ED lymphomas, represented by 47% (8/17) primary lymphomas and 53% (9/17) multicentric. The thoracolumbar segment was most affected (10 cases), including one cat (#79) with multifocal meningeal involvement of the brain and thoracic spinal cord. Canine ED lymphoma was recorded in 9/14 (64.3%) of all canine SCLs. A total of 33.3% (three out of nine) were primary, while 66.7% (six out of nine) were multicentric.

The lesions were usually non-encapsulated, poorly defined, soft, grayish masses within the epidural fat, often with secondary severe spinal cord compression (Figure 5A). In some cases, the tumor infiltrated and invaded the adjacent vertebral bodies or soft tissues. The meninges and spinal cord parenchyma were focally invaded. Most of the diffuse neoplastic infiltration in the epidural fat was composed of large, rounded to oval lymphoblastic cell sheets (Figure 5B). B-cell lymphoma was the most frequently observed phenotype in primary types, both in dogs (three out of three; 100%) and in cats (six out of eight; 75.0%). As for secondary extradural lymphomas in dogs, four out of six (66.7%) were B-cell and two out of six (33.3%) were T-cell lymphomas. In cats, five out of nine (55.6%) were B-cell and four out of nine (44.4%) were T-cell.

Intradural-extramedullary lymphoma was observed in only three (3/21; 14.3%) feline cases, either as a primary or multicentric form. They were all T-cell lymphomas located in the thoracolumbar segment. One of them was associated with a cerebral meningeal mass (#77). In one dog (#35), diffuse intradural-extramedullary neoplastic infiltration was observed throughout the entire spinal cord, showing a pattern consistent with LL.

Intramedullary lymphoma was recorded in 4/14 (28.6%) of canine cases. Two intramedullary tumors were primary lymphoma B-cell and non-B non-T cell phenotypes, and two multicentric lymphomas showed sciatic nerve involvement and lumbosacral intra-medullary invasion (#5 and #9). A primary, non-B non-T cell lymphoma was observed in only one cat (#65).

### 3.3. Peripheral Nervous System Lymphoma

Lymphoma of the peripheral nervous system (or neurolymphomatosis, NL) was observed in 13/45 dogs (28.9%), in two of which it was associated with an intracranial mass and one with concurrent spinal cord neoplasia. Of these cases, 7/13 (53.8%) were males and 6/13 (46.2%) were females. The mean age of dogs with NL was 7.4 ± 3.1 years (ranging from 2.5 to 14 years). There was no breed prevalence: four dogs were mixed breed, and the remaining dogs belonged to the following breeds: Great Dane, Hound, Golden Retriever, Bulldog, Staffordshire Terrier, Amstaff, Cane Corso, Rottweiler, and Labrador Retriever. Eight out of 13 cases (61.5%) were primary lymphomas, while the remaining 5/13 (38.5%) were multicentric. As for neurolocalization, in 7/13 (53.8%) cases the tumor involved the sciatic nerve and/or its branches, in 4/13 (30.8%) cases the spinal roots, spinal nerves and spinal ganglia (i.e., L1, L5, L6 nerves and one case multiple nerves), while in the remaining two cases (15.4%) it involved the trigeminal nerves (Figure 6A,B). Histopathologically, NL was characterized by widespread infiltration of malignant lymphoid cells within the endoneurium and perineurium (Figure 6C). By immunohistochemistry, 6/13 (46.2%) cases were T-cell lymphomas and 7/13 (53.8%) were B-cell lymphomas.

In the feline cases, NL was detected in 7/47 (14.9%) cases, two of which were associated with an intracranial mass. Of these cases, three out of seven (42.9%) were males and four out of seven (57.1%) were females. The age of cats with NL varied widely, ranging from 3 months to 14 years; there was no breed prevalence, as four cats were European Shorthair and the remaining were a Ragdoll, a Carthusian, and an ELH cat. Five out of seven cases (71.4%) were primary lymphomas. As regards the localization, in two cases (#82 and #89) the neoplasm involved both the cranial and spinal nerves. Looking at each compartment individually, in three cases the neoplasm involved the sciatic plexus and/or its branches, and the same occurrence was recorded for spinal roots, spinal nerves, and spinal ganglia. In two cases, NL was observed in cranial nerves, and in another two cases the brachial plexus and/or its branches were involved. By immunohistochemistry, four out of seven (57.1%) cases were B-cell and three out of seven (42.9%) were T-cell lymphomas.

## 4. Discussion

In our study, nervous system lymphoma occurred at any age in dogs and cats, from 3 months to 15 years, but was most common in middle-aged animals, as previously described [5,8,18,26]. Similarly, our results support that there is no breed or gender predisposition, except in dogs, where five (11%) Labrador Retrievers were affected. Neurological signs were present in each patient and were highly variable, depending on the anatomical location of the tumor [20,21]. Both peripheral and central nervous system involvement were identified, and nine different pathological patterns of nervous system lymphoma (intraparenchymal mass, lymphomatosis cerebri, intravascular lymphoma, meningeal mass, leptomeningeal lymphomatosis, extradural, intradural-extramedullary, intramedullary, and neurolymphomatosis) were recorded and will be discussed in this section.

### 4.1. Intracranial Lymphoma

Intracranial lymphoma can involve the brain parenchyma and/or meninges as focal, multifocal, diffuse, and intravascular lesions, showing different histopathological patterns.

#### 4.1.1. Intraparenchymal Patterns

Intraparenchymal lymphomas (IPLs) are commonly reported in feline nervous system lymphomas [2,5,8,24,35,36,37,39,40,46,47,48] and less frequently in the canine counterpart as either primary or multicentric types [3,24,25,26,28,48]. There are currently insufficient data to support a different localization between primary and multicentric tumors; therefore, a thorough post-mortem examination remains mandatory to distinguish these two forms [2]. Rostrotentorial regions such as the cerebral cortex [2,25,36,48,49], diencephalon [1,5,8,28,37], and the olfactory bulb [5,8,48] are the most targeted brain sites. Although considered rarer, cerebellar location has also been described [1,2,4,20,27,40,50].

In our study, canine IPL was observed in the occipital lobe, ponto-cerebellar angle, basal nuclei, thalamus, and parieto-occipital lobe with trigeminal nerve involvement. In cats, no preferred sites were identified, with both the forebrain and hindbrain regions involved. IPLs were mainly multicentric, and no breed or gender predisposition was identified in both species. The mean age of the dogs of our study was 6.8 years, in accordance with previously reported data (from 5.5 to 7.4 years) [1,48]. As noted in this study, where the mean age at diagnosis was 10 years, affected cats are usually reported to be adult or elderly [2,37,48,51].

The distribution of neoplastic cells and the invasion of the neuroparenchyma were similar to those reported in the literature [24,25,28,49]. Lymphoma is usually composed of heterogeneous soft tissue with a whitish-gray appearance and is occasionally associated with small malacic foci [36,49]. Adjacent tissue may be swollen, and multifocal lesions are described [5,48]. Both T-cell [2,24,26,28] and B-cell [8,24,25,35,36,37,40,46] CNSLs have been described in the literature as in this study.

#### 4.1.2. Lymphomatosis Cerebri

The term “lymphomatosis cerebri” (LC) is used to describe an atypical neuroanatomical pattern of lymphoma cell spread [52]. This variant is not so uncommon in the feline species [5,49]. The absence of a cohesive mass is the most notable histological feature of LC. Lymphoid cells spread widely and diffusely mainly within the cerebral white matter from the frontal lobe to the brainstem and spinal cord [3,49,53,54]. Focal leptomeningeal congestion and dilation of the lateral and third ventricles and central canal have also been described [42,54]. Feline LC is always described as a T-cell primary form, although large granular T-cell lymphoma, presumably originating in the alimentary tract with a secondary diffuse infiltrate in the white matter of the brain, has been reported [43]. No breed, age or gender predisposition has been identified [42,54]. In human medicine, no differences in the distribution ratio between sexes have been reported and B-cell LC account for most cases [55,56].

There are no reports of LC in dogs in the veterinary literature. In our case series we described a novel finding of a primary and diffuse infiltration of neoplastic T-cells within the white matter and leptomeninges in one dog that was consistent with LC. No cases of LC in cats were observed in our study.

#### 4.1.3. Intravascular Lymphoma

Intravascular lymphoma (IVL), also known as “angiotropic lymphoma”, is a rare type of lymphoma that has been reported in both humans [57,58] and animals [26,29,34,45,59,60]. This form is characterized by the predominant growth of neoplastic cells within the lumen of blood vessels with little or no extension into the neuroparenchyma. Progressive occlusion of blood vessels with neoplastic cells leads to thrombosis, hemorrhage, and infarction. Various organs could be affected [61,62], but nervous system involvement is the most frequently reported in the literature, both in humans and domestic animals [26,29,34,45,49,60,63]. In dogs, IVL has been described in 10 reports including a total of 51 dogs [26,29,34,45,59,60,63,64,65,66]. In those IVL cases, there was either a restricted involvement of the brain or it was part of a multiorgan/multicentric IVL form. In our study, 10 cases of IVL were observed, representing almost half of the canine intracranial lymphomas. Affected animals were middle-aged dogs with a mean age of 8 years (ranging from 2.5 to 13 years), slightly above what was previously described in two studies (mean age of 7.25 and 6, respectively) [34,59]. In accordance with the literature, no breed or sex predilection was observed in our cases. Although reported in previous studies [29,34,59,66], no spinal cord involvement was detected in our IVL cases, where the lesions were restricted to the brain. The histopathological lesions observed in the affected brain regions were similar to the ones described in previous studies on canine IVL [29,34,59,66]. In our cases, the non-B non-T phenotype was identified in four IVL cases with a prevalence similar to that reported in a previous study [59]. Although the T-cell IVL phenotype was the most prevalent in previous studies [29,59,63], in our remaining six cases there was an equal distribution of T-cell and B-cell IVL.

There were four reports of IVL in cats [8,61,67]. In two of them, the vascular lesions were limited to the brain [8], in one case to the brain and kidneys [67], and in only one case as systemic disease. No gender or breed predisposition for feline IVL was reported in those studies. No cases of feline IVL were recorded in our series.

#### 4.1.4. Extraparenchymal Patterns

Extraparenchymal (or extra-axial) lymphoma (EAL) may appear as a well-defined focal mass involving the meninges but may also diffusely invade the leptomeninges and choroid plexuses, resulting in “leptomeningeal lymphomatosis” and “lymphomatous choroiditis”, respectively [8]. Meningeal masses (MM), unlike intraparenchymal ones, are usually well-defined. They may be irregularly shaped with a whitish-gray appearance and soft texture. Sporadic malacic and hemorrhagic foci are mentioned [49,68].

In this report, intracranial extraparenchymal lymphomas in cats were mainly characterized by meningeal location, usually accompanied by secondary infiltration of the neuroparenchyma. This is a frequent condition in cats, accounting for 30 to 100% of intracranial lymphomas [5,18,21,48], as a primary or multicentric type [48]. Feline EAL was diagnosed more commonly than IPL in our series, primarily as primary types. This pathological pattern is less frequent in dogs and has been reported in a few cases [31,48]. Additionally, in our series only two dogs had an occipito-temporal meningeal lymphoma and a multifocal meningeal infiltration also involving the thalamus, cerebellum, medulla oblongata, choroid plexus, and pituitary gland.

In cats, most tumors described by Mandara et al. (2022) [8] were within the cranial fossa. Interestingly, all feline MM in our study were in the forebrain, with the frontal lobe being the most common site. The median age of cats was 11 years (ranging from 4 to 17), with ESH as the main represented breed in accordance with the existing literature [5,8,38]. Based on the immunohistochemistry results, most intracranial meningeal masses in our feline cases were B-cell lymphomas, as reported in previous studies [8,38].

EAL may also develop in the choroid plexus (CP) and either grow to a recognizable mass [27,68] or lead to diffuse neuroparenchymal infiltration [8,18,26,30]. In both human and veterinary medicine, the latter variant, known as “lymphomatous choroiditis,” is sporadically documented [27,68].

In our series, four and two cases with CP involvement were observed in dogs and cats, respectively. However, neoplastic lesions within the CP were always associated with lymphoma spread to other CNS or PNS locations, both as primary and multicentric forms.

An additional neuroanatomical growth pattern of extraparenchymal lymphoma is intradural lymphoma. In humans, an uncommon subtype of primary CNSL that develops from the dura mater and differs considerably from other CNSLs is called primary dural lymphoma. This tumor is classified as a mucosa-associated lymphoid tissue (MALT) lymphoma and is typically indolent [69,70]. However, secondary involvement of the dura and leptomeninges is reported [71]. In veterinary medicine, pachymeningeal lymphoma is poorly documented, and Mello et al. [18] reported the lone occurrence of intradural lymphoma in cats. In our case series this pathological pattern was not observed.

#### 4.1.5. Leptomeningeal Lymphomatosis

When neoplastic lymphocytes show prevalent diffuse infiltration of the subarachnoid space, this pathological pattern is referred to as leptomeningeal lymphomatosis (LL), or lymphomatous meningitis, and has been reported in dogs and cats [8,49,72]. In some reports it has been considered as a secondary evolution of an extra-nervous lymphoma [32,72], or as a primary form [22,23,30,41,73]. Neoplastic lymphocytes are thought to directly extend from pre-existing primary or metastatic CNS tumors, from an extra-CNS mass invading the PNS, or through hematogenous dissemination into the leptomeningeal structures [72,74]. However, since primary forms have been reported, the etiopathogenesis of LL remains unknown. Being frequently described alongside neoplastic infiltrates occurring in the nervous tissue, such as neurolymphomatosis [2,8] or parenchymal and periventricular infiltrates [43,54], this form represents a diagnostic challenge [72].

In the present study, two cases of canine LL (2/45; 4.4%) were detected as a multicentric form: one was found in the cerebral and cerebellar lobes, and the other involved the meninges of the entire spinal cord. Canine LL is reported to be composed of T-cell [30,31] or B-cell [23,26] phenotypes. Both T-cell and B-cell phenotypes were represented in our study.

Recently, Mandara et al. (2022) [8] described three cases of primary feline LL: one case with involvement limited to the brain, one to the spinal cord, and one with a pathological continuum between brain LL and neurolymphomatosis of the optic chiasm, all affecting DSH adult/elderly cats. Feline LL was also described in primary [2] and in secondary brain lymphoma [18] as the most frequent pathological intracranial pattern. In our study, two cases of LL were observed in cats with occasional leptomeningeal thickening. One of these was primary and limited to the brain meninges, while the second was a multicentric type affecting both the brain and spinal cord.

The feline primary form described in our series was T-cell lymphoma. Notably, five out of seven primary LL described in the literature were classified as T-cell lymphoma [2,8,41]. The two B-cell primary LL described in the literature showed rare oculo-cerebral involvement [8,73]. Further studies are needed to assess whether there is an association between the phenotype and this location. Regarding LL as part of multicentric lymphoma in cats, both B-cell and T-cell phenotypes have been reported [17,18,72].

### 4.2. Intraspinal Lymphoma

Lymphoma within the vertebral canal (or intraspinal lymphoma, SCL) can be categorized into three main forms with distinct biological characteristics based on anatomical location within the vertebral canal and spinal cord involvement [7,49]. Extra-axial lesions may be external to the dura mater (extradural; ED) or internal to the dura mater but external to the pia and spinal parenchyma (intradural-extramedullary; ID-EM). The SCL may also be located within the spinal cord parenchyma (intramedullary; IM) [14,75,76,77,78,79].

Most frequently, SCLs are reported as secondary [7,14,18]; however, primary forms are also described [2,7,8,14,17]. The mechanism of primary SCL formation is not fully understood; hematopoietic tissues of extramedullary or vertebral bone have been suggested as the site of origin of spinal lymphoma [7,14,80,81,82]. Spinal cord secondary lymphomas are thought to result from direct extension from the paravertebral regions through the vertebral foramen, or from hematogenous dissemination through the epidural venous system [83].

Canine SCL occurs almost as frequently as in cats. The most common location is extradural [20,22,32,33,48,79,80,81,84,85,86], followed by intramedullary [20,48,75,76], and less commonly intradural-extramedullary location [20,48]. In our study, SCL was more frequently detected in cats than in dogs, with the thoracolumbar location over-represented in both species.

Cats with spinal lymphoma are often younger than those with other spinal cord tumors [7]. In cats, the mean age at the time of diagnosis was highly variable, with 4.5 years (ranging from 8 months to 7 years) and 4 years (ranging from 1 to 11 years) for primary and secondary forms, respectively [14,77,87]. Similarly, age at diagnosis was variable in our study, with a comparable mean age between primary (5 years; range 0.7–12 years) and secondary (6 years; range 0.7–13 years) types. No gender or breed predisposition was found in our study, as in the series reported in the literature.

Clinical and pathological features of SCL may vary. It can manifest as a focal mass [14,32,33,77], multifocal masses, or extensively disseminated tumor [8,14]. The lumbosacral and thoracic segments are the preferential site of onset [32,33], but lymphoma has the potential to involve any segment of the spinal cord [8,14,18,20,79,87]. Furthermore, SCL is often reported in association with cranial [7,88] or spinal nerves [2,21,68,77] involvement, depending on the anatomical location.

#### 4.2.1. Extradural Lymphoma

Our findings on ED lymphoma are comparable to those of previous studies, where ED lymphoma was the most frequently reported pattern of SCL, described as both primary and secondary type [7,14,20,22,32,33,48,68,77,80,81,84,85,86,89]. Both T-cell and B-cell phenotype were found, as previously reported [2,8,17,18,20,32,33,81,90,91]. Most of the primary ED lymphoma in our cats and all primary types in dogs were B-cell lymphomas, whereas the B and T phenotypes were equally represented in the multicentric types. Further studies on feline SCL phenotyping and its possible association with neuroanatomical patterns are warranted. Based on the limited data available, B-cell lymphoma seems to be the most prevalent phenotype in cats. Although numerous cases have been reported in dogs, immunophenotyping was performed in only a few of them [20,32,33,81], showing that the most frequent phenotype was T-cell.

#### 4.2.2. Intradural-Extramedullary Lymphoma

In the present study, only one case of canine ID-EM lymphoma was found in a dog and three in adult cats. ID-EM has been reported in both species [7,20,88] as an irregular mass that fills the subdural and subarachnoid spaces and tends to infiltrate contiguous segments of the spinal cord or nerve roots [49,87]. In two primary intradural-extramedullary lymphomas, T-cell phenotypes were identified [2]. In our case series, all tumors were T-cell, and in one case it was associated with intracranial meningeal involvement.

#### 4.2.3. Intramedullary Lymphoma

Intramedullary pattern is occasionally reported in dogs and cats, mainly as secondary SCL [7,20,76]. The tumor may appear as a soft, poorly defined mass or may produce neuraxis enlargement, altering the parenchymal architecture [2,7]. Hemorrhagic myelomalacia due to compression injury has been documented in cats in both intramedullary and extramedullary lymphomas [91,92].

In our study, four tumors in dogs and only one tumor in a cat were classified as intramedullary SCL. Given the few cases, no differences between primary and secondary forms were identified in both species. The feline tumor was a non-B non-T lymphoma. Notably, no intramedullary B-cell lymphomas have been reported in cats [8,17].

### 4.3. Peripheral Nervous System Lymphoma

Peripheral nerve lymphoma (or neurolymphomatosis, NL) is uncommon in dogs and rare in cats [12,15,51,93]. They can involve any cranial nerve, spinal nerve roots, somatic, or autonomic nerves [78,94,95,96]. The most frequently affected nerves are the trigeminal, those within the cervico-thoracic segment (C6-T2), and, more rarely, nerves of the lumbar intumescence [93]. The neoplasms may be confined to or extend along the nerves, resulting in an intradural-extramedullary location with spinal cord compression [97,98,99]. Multicentric NL is rarely reported in both human and animal species [12,100,101,102,103,104].

Macroscopically, the affected peripheral nerve may be normal or homogeneously thickened [105], soft, and have yellowish discoloration [18], mimicking inflammatory infiltration or the less common chronic hypertrophic neuritis [106]. Less frequently, a mass that focally deforms the nerve trunk can be observed, whereas in other cases, a large tumor mass is easily identifiable [16,107]. These masses are commonly described as solid, whitish, irregularly bosselated, and non-symmetric [103]. Complete nerve loss and replacement with a well-circumscribed white mass has also been reported [16]. In several cases, moderate atrophy of the muscles innervated by the affected nerves was the most significant gross lesion associated with NL [15,16,105,107].

In our study, the higher prevalence in dogs than in cats is in contrast with the data reported in the literature [2,8,9,10,11,12,13,14,15,16,18,101,105,107]. Our results showed no breed or gender predisposition in either dogs or cats and that NL typically affected adults. The mean age of dogs and cats with NL was similar to that reported in the literature. In dogs, the prevalence of the primary type was higher than the secondary, unlike what was reported in other studies, in which all cases were secondary types. Regarding the feline species, NL was primary in most of the cases, accordingly with previous studies [2,8,15,16,18,101]. The increase in the number of primary lymphoma diagnoses may be related to the development of veterinary neurology and neurosurgery referral services [8].

In our dogs, the sciatic nerve and/or its branches was the most affected of the compartments, followed by the spinal roots, spinal nerves and spinal ganglia, and the trigeminal nerve. Similar findings have been reported in the literature [10,11,12,13]. Results on the feline population showed that there was no predominance of one compartment over the others [2,8,14,15,16,18,101,103,105,107]. Optic nerve involvement may be more frequent in cats [8]. Moreover, PNS involvement with and without concurrent CNS involvement are findings similar to those reported in the literature [2,8,9].

Data on phenotype are still fragmentary in veterinary medicine, as the only data are single case reports or small cohorts. Our results showed that there is no predominance of one phenotype over the other, but B- and T-cells lymphomas are equally distributed among population in both dogs and cats. These results are in line with the literature [2,8,14,15,16,18,101,103,105,107].

## 5. Conclusions

In this retrospective study we evaluated 92 cases of central and peripheral nervous system lymphoma in dogs and cats, which were studied on the basis of consistent histopathological and immunohistochemical examination. An extensive literature review was performed to collate available information on reported cases and reconcile fragmented clinical and pathological data so that our results could be compared with the available literature.

In our study, no age, breed, or gender predisposition related to specific CNSL subtypes was seen in both species, even if in Labrador Retriever dogs it occurred with a slightly higher prevalence. The lone exception was the occurrence of SCL in juvenile cats, as previously reported. The overall anatomical locations of CNSL vs. PNSL, type (primary vs. secondary/multicentric), and phenotypes were similar across the species.

Regarding CNSL, the forebrain was the most frequent site of lymphoma in dogs and the thoracolumbar segment was the most frequent site in cats. Primary CNSL in cats most frequently involved the forebrain meninges and, to a lesser extent, the optic nerves, especially as a B-cell phenotype. Extradural location was the most prevalent pathological pattern of SCL in both species. The B-cell phenotype predominated in primary lymphoma, whereas the B-cell and T-cell phenotypes were equally represented in the multicentric type.

PNSL most frequently affected the sciatic nerves and its branches in dogs, whereas no prevalent compartment was recognized in cats. B-cell and T-cell phenotypes were equally represented in primary and multicentric PNSL in both species.

Canine intravascular lymphoma mimicked a neurovascular condition rather than space-occupying or inflammatory lesions. Lymphomatosis cerebri can also occur in dogs.

Clinicopathological features were variable in both species, particularly in neurolymphomatosis, due to different combinations of central and peripheral nervous system involvement.

## Figures and Tables

**Figure 1 animals-13-00862-f001:**
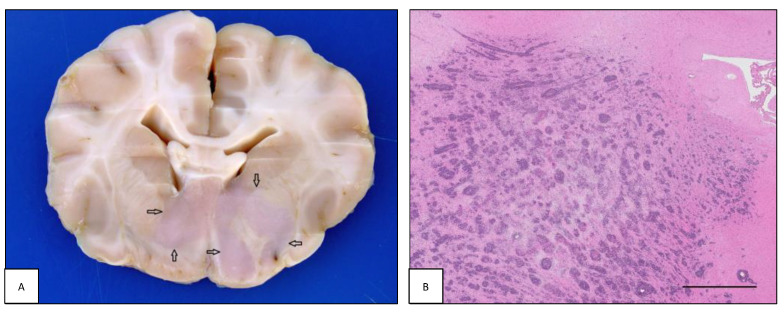
Intracranial lymphoma: intraparenchymal pattern. (**A**) Case #11. Dog brain section at the level of the internal capsule. A focal extensive intraparenchymal lymphoma involves the diencephalic structures (open arrows). (**B**) Case #55. Feline brain section showing perivascular arrangement of intraparenchymal lymphoma with extensive infiltration of the corona radiata and diencephalic structures. Note tissue rarefaction in the central area of the neoplasia (H&E, bar = 2.5 mm).

**Figure 2 animals-13-00862-f002:**
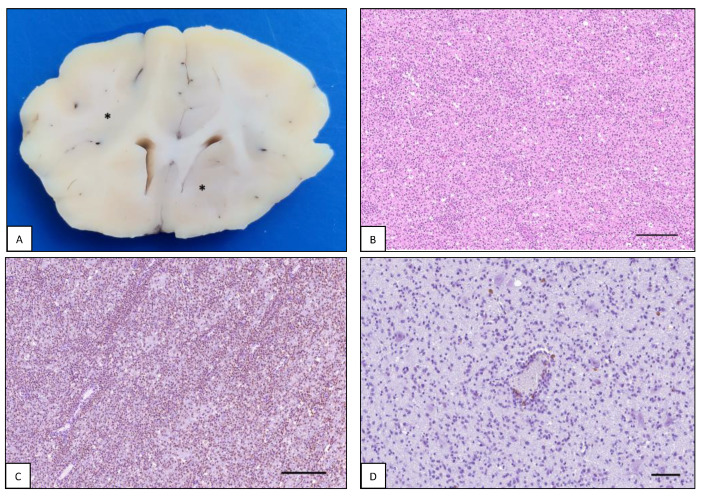
Intracranial lymphoma: lymphomatosis cerebri. Dog brain, case #39. (**A**) Dog brain section at the level of septal and basal nuclei. Bilaterally, large areas of the corona radiata and basal nuclei are grayish discolored (asterisks). The gray to white matter interface is also diffusely blurred. (**B**) Malignant cells diffusely infiltrate the neuroparenchyma (H&E, bar = 250 µm). (**C**) Neoplastic cells are immunoreactive with anti-CD3 antibody (IHC, bar = 250 µm). (**D**) Only a few perivascular and scattered lymphocytes within the surrounding neuroparenchyma are labelled with anti-CD20 antibody (IHC, bar = 150 µm).

**Figure 3 animals-13-00862-f003:**
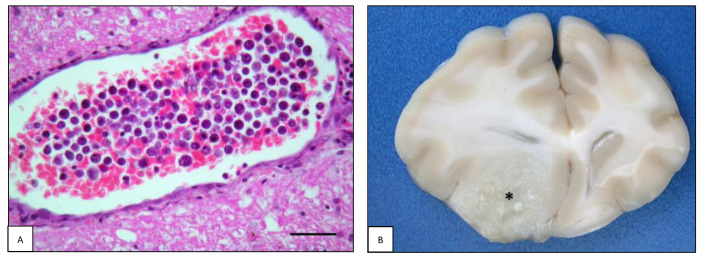
Intracranial lymphoma. (**A**) Intravascular lymphoma. Case #30. Dog brain section with blood vessel lumen filled with large neoplastic cells (H&E, bar = 100 µm). (**B**) Meningeal lymphoma. Case #58. Cat brain section with a focally extensive meningeal lymphoma (asterisk) accompanied by intraparenchymal invasion and perilesional edema causing enlargement of the right hemisphere and left shifting of the midline structures.

**Figure 4 animals-13-00862-f004:**
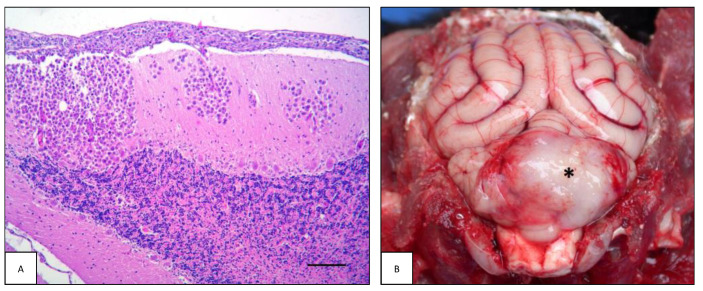
Intracranial lymphoma: leptomeningeal lymphomatosis. (**A**) Case #40. Dog brain section in which neoplastic cells diffusely infiltrate the cerebellar leptomeninges and multifocally invade the molecular layer (H&E, bar = 100 µm). (**B**) Case #49. Cat brain, the surface of the cerebellum is covered by a whitish and thick layer histologically composed of neoplastic cells (asterisk).

**Figure 5 animals-13-00862-f005:**
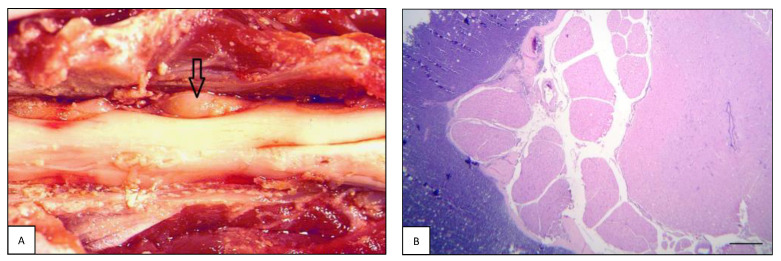
Feline intraspinal lymphoma. (**A**) Case #67. The thoracic spinal cord is focally compressed by an extradural lymphoma (open arrow). (**B**) Case #53. Lumbar spinal cord. Extradural accumulation of neoplastic lymphocytes with only minimal infiltration of the dura mater (H&E, bar = 1 mm).

**Figure 6 animals-13-00862-f006:**
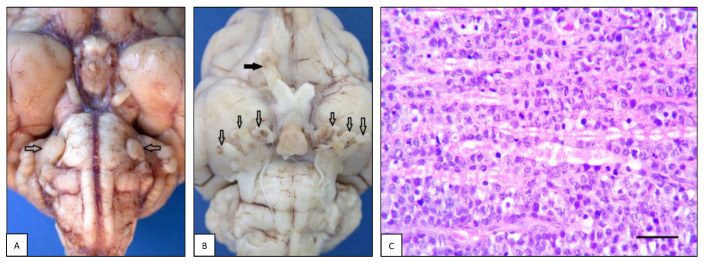
Peripheral nervous system lymphoma: neurolymphomatosis. (**A**) Case #21. Dog brain, ventral view. Both roots of the trigeminal nerve are enlarged due to infiltration of neoplastic cells (open arrows). (**B**) Case #89. Cat brain, ventral view. Ventral aspect of cat brain with neurolymphomatosis. All three branches of both trigeminal nerve roots (open arrows) and right optic nerve (black arrow) are enlarged due to infiltration of neoplastic cells. (**C**) Case #9. Canine sciatic nerve longitudinal section. Neoplastic cells extensively infiltrate the endoneurial tissue. Nerve fibers show diffuse myelin degeneration (H&E, bar = 100 µm).

**Table 1 animals-13-00862-t001:** Signalment, main neurological signs, and tissue sample origin of 92 cases of canine and feline central and peripheral nervous system lymphoma.

Case#	Species	Breed	Sex	Age	Neurological Signs	Tissue Origin
1	cn	Border Collie	F	2 y	no menace response, nystagmus, trigeminal paralysis	necropsy
2	cn	Mixed breed	M	10 m	paraparesis, back pain	biopsy
3	cn	German Shepherd	F	8 y	acute paraparesis	necropsy
4	cn	Cane Corso	SF	5 y	ataxia, depression, strabismus, left hemiparesis	necropsy
5	cn	Great Dane	M	2.5 y	monoparesis, cauda equina syndrome	necropsy
6	cn	Mixed breed	F	6 y	monoparesis	biopsy
7	cn	Cane Corso	M	1 y	paraparesis	biopsy
8	cn	Epagneul Breton	M	2 y	absent menace response, mydriasis, ataxia, head tilt	necropsy
9	cn	Hound	F	7 y	progressive weakness, ataxia, ascending paresis	necropsy
10	cn	Golden Retriever	M	7 y	multifocal intracranial syndrome	necropsy
11	cn	Shih-tzu	M	8 y	ataxia, tetraparesis, no menace response, absence of pupillary light reflex	necropsy
12	cn	Bulldog	M	4 y	right HL monoplegia	biopsy
13	cn	Mixed breed	SF	7 y	acute and progressive paraparesis	biopsy
14	cn	West Highland White Terrier	M	4 y	forebrain syndrome, blindness	necropsy
15	cn	Labrador Retriever	NM	12 y	paraplegia	necropsy
16	cn	Labrador Retriever	M	6 y	tetraparesis	necropsy
17	cn	Yorkshire Terrier	M	7 y	paraparesis	necropsy
18	cn	Border Collie	SF	13 y	right head tilt, brainstem syndrome	necropsy
19	cn	Beagle	M	7 y	acute paraparesis	necropsy
20	cn	Mixed breed	M	10 y	cervical pain, tetraparesis	necropsy
21	cn	Staffordshire Terrier	F	6.5 y	↓ menace response, masticatory muscles atrophy, ↓ facial sensitivity	necropsy
22	cn	Labrador Retriever	F	3 y	left HL monoplegia, right HL monoparesis	necropsy
23	cn	Mixed breed	M	4.5 y	paraparesis	biopsy
24	cn	Rottweiler	M	2 y	depression, ataxia, no menace response, anisocoria, ↓ facial sensitivity	necropsy
25	cn	Labrador Retriever	M	7 y	forebrain syndrome	necropsy
26	cn	Bergamasco Shepherd	M	7 y	paraparesis, severe depression	necropsy
27	cn	Boxer	F	3.5 y	forebrain syndrome	necropsy
28	cn	English Cocker Spaniel	M	11 y	forebrain syndrome	necropsy
29	cn	Boxer	F	4 y	intracranial multifocal syndrome, seizures	necropsy
30	cn	Pointer	M	13 y	paraparesis, depression	necropsy
31	cn	German Shepherd	F	9 y	tetraparesis, depression	necropsy
32	cn	Mixed breed	SF	12 y	right hemiparesis	necropsy
33	cn	Dachshund	F	2.5 y	depression, seizures	necropsy
34	cn	Boxer	F	10.5 y	seizures	necropsy
35	cn	Mixed breed	F	4 y	tetraplegia	necropsy
36	cn	Mixed breed	M	14 y	left HL monoparesis	necropsy
37	cn	Mixed breed	M	9 y	tetraparesis	necropsy
38	cn	German Shepherd	F	3 y	paraparesis	necropsy
39	cn	Hound	F	2 y	severe depression, tetraparesis	necropsy
40	cn	Boxer	F	7 y	neck pain, intracranial syndrome	necropsy
41	cn	Mixed breed	M	14 y	central vestibular syndrome	necropsy
42	cn	Amstaff	M	3 y	right HL progressive lameness	biopsy
43	cn	Cane Corso	F	7 y	monoplegia	biopsy
44	cn	Rottweiler	F	4.5 y	right HL monoparesis	biopsy
45	cn	Labrador Retriever	M	10 y	tetraparesis, paraplegia	biopsy
46	fn	ESH	F	9 y	bilateral blindness, depression, seizures,	necropsy
47	fn	ESH	F	5 y	ataxia, left HL monoparesis, strangury	biopsy
48	fn	Persian	M	11 y	seizures, ataxia, blindness	necropsy
49	fn	ESH	M	6 y	progressive paraparesis, ataxia	necropsy
50	fn	ESH	F	14 y	ataxia, depression	necropsy
51	fn	ESH	M	9 y	ataxia, depression, vertical nystagmus	biopsy
52	fn	ESH	NM	14 y	depression, compulsive gait	necropsy
53	fn	ESH	M	11 y	paraplegia	necropsy
54	fn	ESH	M	8 y	forebrain syndrome	biopsy
55	fn	ESH	NM	17 y	seizures, depression, tetraparesis, bilateral mydriasis	necropsy
56	fn	ESH	M	8 y	left FL monoplegia and muscle atrophy	necropsy
57	fn	ESH	SF	5 y	paraplegia	biopsy
58	fn	ESH	F	12 y	forebrain syndrome	necropsy
59	fn	ESH	M	3 y	paraplegia	necropsy
60	fn	ESH	M	1 y	back pain, paraparesis	biopsy
61	fn	ESH	F	10 y	prosencephalic syndrome	necropsy
62	fn	ESH	NM	11 y	depression, right drifting, compulsive gait	biopsy
63	fn	ESH	NM	12 y	compulsive gait, left circling	necropsy
64	fn	ESH	NM	4 y	seizures	necropsy
65	fn	ESH	SF	12 y	progressive tetraparesis	necropsy
66	fn	ESH	M	11 y	left blindness, forebrain syndrome	necropsy
67	fn	ESH	M	4 y	paraparesis	necropsy
68	fn	ESH	SF	5 y	progressive paraparesis	biopsy
69	fn	ESH	SF	15 y	blindness, forebrain syndrome	necropsy
70	fn	ESH	SF	8 y	ataxia, behavioral changes, mydriasis, left pleurothotonus	necropsy
71	fn	ESH	NM	9 y	brainstem syndrome	necropsy
72	fn	ESH	SF	4 y	paraplegia	biopsy
73	fn	ESH	NM	1 y	paraplegia, loss of deep pain sensation	biopsy
74	fn	Ragdoll	F	3 m	seizures	necropsy
75	fn	ESH	SF	3 y	paraplegia	necropsy
76	fn	ESH	M	10 y	paraplegia, back pain	necropsy
77	fn	Maine Coon	F	13 y	depression, paraparesis	necropsy
78	fn	ESH	SF	8 m	paraparesis	biopsy
79	fn	ESH	M	8 m	paraplegia, loss of deep pain sensation	necropsy
80	fn	Norwegian Forest cat	SF	11 y	depression, weakness, ↓ left menace response and left pupillary light reflex	necropsy
81	fn	ESH	F	12.5 y	left circling, anisocoria, left miosis, right FL monoparesis	necropsy
82	fn	Carthusian	M	6 y	progressive paraparesis	necropsy
83	fn	ESH	SF	10 y	paraparesis	necropsy
84	fn	ESH	F	2 y	acute paraplegia	necropsy
85	fn	ESH	SF	11 y	tetraparesis	necropsy
86	fn	Siamese	SF	7 y	paraparesis, hyperesthesia, ↓ pain sensation, ataxia, urinary loss	necropsy
87	fn	ESH	F	7 y	severe paraparesis	biopsy
88	fn	ESH	M	14 y	right hemiparesis	necropsy
89	fn	ESH	SF	7 y	acute tetraplegia, dysphagia, right Horner syndrome	necropsy
90	fn	ESH	M	14 y	progressive tetraparesis	biopsy
91	fn	ELH	SF	12 y	right HL monoplegia	biopsy
92	fn	ESH	SF	11 y	lameness and monoparesis	biopsy

cn = canine; fn = feline; ESH = European Shorthair cat; ELH = European Longhair cat; M = male; NM = neutered male; F = female; SF = spayed female; y = years; m = months; FL = forelimbs; HL = hindlimbs; ↓ = reduced.

**Table 2 animals-13-00862-t002:** Type, anatomical site, location, distribution, pathological pattern, nuclear size, grading, and phenotype of nervous system lymphoma in 92 canine and feline cases.

N°	Species	Type	Anatomical Site	Location and Distribution	Pathological Pattern	Nuclear Size	Grading	Phenotype
1	cn	II	IC	occipital lobe	IP	large	medium	B
2	cn	I	SC	T10-T13	ED	intermed.	low	B
3	cn	II	SC	L3-L5	ED	large	high	B
4	cn	II	IC	occipito-temporal lobe	IP, MM	large	high	T
5	cn	II	SC, PNS	cauda equina, sciatic nerve	IM, NL	small	low	B
6	cn	I	PNS	L6 nerve root	NL	large	high	B
7	cn	I	SC	T9	ED	large	high	B
8	cn	II	IC	occipital lobe	IP	large	high	non-B, non-T
9	cn	II	SC, PNS	LS, sciatic nerves	IM, NL	large	high	T
10	cn	I	IC, PNS	parieto-occipital lobes, V nerve root	IP, NL	large	high	B
11	cn	II	IC	basal nuclei	IP	large	high	T
12	cn	II	PNS	ischiatic nerve	NL	large	high	B
13	cn	II	PNS	L1 nerve root	NL	large	high	B
14	cn	II	IC	occipital lobe	IP	large	high	B
15	cn	II	SC	L	ED	large	high	B
16	cn	II	SC	C3-C6	ED	large	medium	B
17	cn	II	SC	T11-T13	ED	small	low	T
18	cn	II	IC	thalamus, cerebellum, m. obl., CP, PG	IP, MM	large	high	B
19	cn	I	SC	T6-T7	IM	large	high	non-B, non-T
20	cn	I	SC	C3-C4	IM	intermed.	low	B
21	cn	I	IC, PNS	trigeminal nerves, CP	NL	intermed.	low	B
22	cn	I	SC	L4-L7	ED	large	high	B
23	cn	II	SC	T12-L1	ED	large	high	B
24	cn	I	IC	thalamus	IP	large	high	B
25	cn	I	IC	multifocal	IVL	large	ND	B
26	cn	I	IC	multifocal	IVL	large	ND	non-B, non-T
27	cn	I	IC	multifocal	IVL	large	ND	T
28	cn	I	IC	multifocal, CP	IVL	large	ND	non-B, non-T
29	cn	I	IC	multifocal	IVL	large	ND	B
30	cn	I	IC	multifocal, CP	IVL	large	ND	B
31	cn	I	IC	multifocal	IVL	large	ND	T
32	cn	II	IC	multifocal	IVL	large	ND	non-B, non-T
33	cn	I	IC	multifocal	IVL	large	ND	T
34	cn	I	IC	multifocal	IVL	large	ND	non-B, non-T
35	cn	II	SC	C, CT, TL, LS	ID-EM, LL	small	medium	B
36	cn	I	PNS	L femoral nerve	NL	large	high	T
37	cn	II	PNS	multiple spinal nerves	NL	intermed.	low	T
38	cn	II	SC	T	ED	small	medium	T
39	cn	I	IC	diffuse	LC	intermed.	medium	T
40	cn	II	IC	frontal, temporal, cerebellar lobes	LL	large	high	T
41	cn	II	IC	ponto-cerebellar angle	IP	large	high	B
42	cn	I	PNS	L5 nerve root	NL	large	low	B
43	cn	I	PNS	sciatic nerve	NL	large	low	T
44	cn	I	PNS	sciatic nerve	NL	large	medium	T
45	cn	I	PNS	sciatic nerves	NL	small	low	T
46	fn	I	IC	optic nerves, diencephalon	IP	intermed.	medium	B
47	fn	I	SC	L4-L5	ED	intermed.	low	B
48	fn	II	IC	parietal lobe	IP	intermed.	medium	T
49	fn	II	IC, SC	cerebellar, T	LL, ED	large	medium	B
50	fn	I	IC	frontal lobe	IP, MM	large	high	non-B, non-T
51	fn	I	IC	frontal lobe	IP, MM	large	high	B
52	fn	II	IC	frontal lobe, CP	IP, MM	large	high	B
53	fn	II	SC	LS	ED	small	medium	T
54	fn	I	IC	parietal lobe	IP, MM	large	high	B
55	fn	I	IC	diencephalon, thalamus	IP, MM	small	low	T
56	fn	II	PNS	brachial plexus	NL	large	medium	T
57	fn	I	SC	T12-L1	ED	small	low	B
58	fn	I	IC	frontal lobe, diencephalon	IP, MM	large	high	T
59	fn	I	SC	L5-L6	ED	large	medium	B
60	fn	II	SC	L1	ED	large	high	B
61	fn	II	IC	frontal lobe	MM	large	medium	non-B, non-T
62	fn	I	IC	frontal lobe	IP	intermed.	medium	T
63	fn	I	IC	frontal lobe	IP, MM	large	high	B
64	fn	I	IC	frontal lobe	IP, MM	large	high	B
65	fn	I	SC	CT	IM	large	high	non-B, non-T
66	fn	II	IC	diencephalon	IP, MM	intermed.	high	B
67	fn	II	SC	T8	ED	large	high	B
68	fn	I	SC	L3	ED	large	medium	B
69	fn	II	IC	diencephalon	IP, MM	large	high	B
70	fn	II	IC	medulla oblongata	IP	intermed.	low	B
71	fn	II	IC	medulla oblongata	IP	large	low	T
72	fn	I	SC	L3-L4	ED	small	medium	B
73	fn	II	SC	T11	ED	small	medium	T
74	fn	I	PNS	cervical spinal nerves	NL	large	low	B
75	fn	I	SC	TL	ED	small	low	T
76	fn	II	SC	TL	ID-EM	intermed.	low	T
77	fn	II	IC, SC	cerebral cortex, m. obl., T	ID-EM, MM	small	medium	T
78	fn	I	SC	T5-T7	ED	intermed.	medium	B
79	fn	II	IC, SC	TL, CP	MM, ED	small	low	B
80	fn	I	IC	olfactory lobe	IP, MM	large	high	B
81	fn	I	IC	parietal lobe	IP, MM	intermed.	medium	T
82	fn	II	IC, PNS	C and TL nerve roots	NL	small	low	B
83	fn	II	SC	TL	ED	small	low	T
84	fn	II	SC	TL	ED	small	low	B
85	fn	II	SC	CT	ED	intermed.	medium	T
86	fn	I	SC	LS	ED	intermed.	low	non-B, non-T
87	fn	I	SC	T11-T12	ID-EM	large	medium	T
88	fn	I	IC	forebrain, midbrain	LL	large	high	T
89	fn	I	IC, PNS	II, V cranial, brachial, sciatic nerves	NL	intermed.	low	B
90	fn	I	PNS	C7 nerve root	NL	small	low	T
91	fn	I	PNS	sciatic nerve	NL	small	medium	T
92	fn	I	PNS	sciatic nerve	NL	large	medium	B

cn = canine; fn = feline; I = primary form; II = secondary/multicentric form; IC = intracranial; SC = intraspinal; PNS = peripheral nervous system; C = cervical; CT = cervicothoracic; T = thoracic; L = lumbar; TL = thoracolumbar; S = sacral spinal cord; CP = choroid plexus; PG = pituitary gland; IP = intraparenchymal mass; MM = meningeal mass; LL = leptomeningeal lymphomatosis; IVL = intravascular lymphoma; LC = lymphomatosis cerebri; ED = extradural; ID-EM = intradural-extramedullary; IM = intramedullary; NL = neurolymphomatosis, ND = not determined.

## Data Availability

Not applicable.

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
