# Peer review of "Neuropathology of Central and Peripheral Nervous System Lymphoma in Dogs and Cats: A Study of 92 Cases and Review of the Literature"

_animals, 2023, doi:10.3390/ani13050862_

Round 1

Reviewer 1 Report

The article describes 92 cases of canine and feline nervous lymphomas with a review of litterature on this topic.

This is an extensive cohort, tumors are well characterized with histopathology and immunohistochemestry available for all cases. The authors also describe the first case of canine lymphomatosis cerebri. The review of literature is very complete, references include the majority of, if not all, recent publications on this topic, in internationally recognized journals.  The article is well written, results are clearly presented.  Authors make a considerable contribution to characterization of nervous system lymphoma in companion animals.

Specific comments:

Lines 38-40: “Primary CNSL is a relatively uncommon form of tumor in dogs and rare in cats, accounting for […] less than 3% in cats where primary CNS infiltration is observed”.  The sentence seems unclear to me. I would suggest  “less than 3% of cats with primary CNS tumors.”

Lines 82-83:  “Extra neural system involvement was assessed clinically or after complete post-mortem examination in case of death”. Would you be able to quantify in which percentage of cases a complete post-mortem examination was available?  This would be relevant because assessment of extra-neural involvement could be considered as a weakness owing to the retrospective design (without systematic complete staging).

Reviewer 2 Report

Comments for the Authors Manuscript Animals 2227841

The manuscript paper by Fonti et al. is a good contribution to veterinary science on the subject of nervous system lymphomas in dogs and cats as it is scattered in the literature with no cohesion. The authors do a very good job of coalescing this topic with an extensive and updated bibliography carrying out at the same time a retrospective analysis in 92 cases of canine and feline nervous system lymphomas.

In general, the results are similar to that described in the veterinary literature but two differences are highlighted in the present study:  

  1. the description for the first time of lymphomatosis cerebri (T-cell primary form) in dogs with a dedicated figure (Figure 2);
  2. the high prevalence of neurolymphomatosis in dogs than in cats, occurring predominantly as primary type in the sciatic nerve and/or its branches, even though the mean age and phenotype were in line with the literature.

The authors also emphasize the importance to get sistematic data on lymphoma phenotypes to assess if there is an association between the phenotype, pattern and location, especially in the case of rare neuroanatomical distributions. 

As the authors did a comprehensive literature search, they also should emphasize what they have done regarding that review and its tie-in with what they found with the retrospective study in the first paragraph of the Conclusion.

I think that the following points should be also addressed to clarify some issues:

·         As there is a multitude of abbreviations it would be convenient to include a Table of Abbreviations to facilitate the reading.

 Page 1: 

  • Lines 14 and 21- Meaning of “heterogeneous results” – different, divergent or assorted?  
  • Line 29- Please introduce “central nervous system lymphoma” before the abbreviation “CNSL”; 

Page 2 

  • Line 86- Please include the thickness of the tissue sections; 
  • Lines 88-90 – Please complete the information about the antibodies, namely if they are B-cell or T-cell markers, considered diagnostic for B cell and T cell lymphoma, clarifying the readers not used to immunohistochemistry (ex. … anti-human-CD20, …. anti-human CD3, …..anti-human-Paried Box 5 (PAX5)). In this panel of antibodies why the authors did not include the mouse monoclonal anti-human-multiple myeloma 1 (MUM-1) and mouse monoclonal anti-human-CD79a, antibodies considered important for lymphoma phenotyping?;
  • Line 91 Please introduce “phosphate buffered saline” before the abbreviation “PBS”;
  • Line 93 Please confirm if there was no antigen retrieval treatment for instance for anti-CD3 and Pax5;

Page 3 

  • Line 95 Please include the incubation time for the secondary antibody;
  • Line 102 Please include the criteria of the scoring system used to assess the percentage of positive neoplastic cells for the lymphoma phenotyping, despite these results are not described; in my opinion, these immunohistochemical findings should be stated together with the phenotype, for instance in Table 2;
  • In the Materials and Methods why the authors did not consider evaluating the histological grade of the lymphomas based on the mitotic count? Even in line 184, the authors express the number of mitotic figures in the description of case #39 that showed a pattern consistent with lymphomatosis cerebri (LC) that is described for the first time in a dog by the authors; this first report should be fully characterized;
  • Table 1- It is suggested to include information on how the case was analysed (Biopsy or necropsy);
  • Line 116 the pathological pattern lymphomatosis cerebri (LC) is not described in this Section 2.3 and appears in table 2, becoming an important observation and result in this manuscript;

Page 7

·         Table 2- It is suggested to add the immunohistochemistry results in each case or include another table with this characterization and eventual histological grade;

Page 8

·         Line 144- It is suggested to replace “9/45 (20%) affected the peripheral nervous system” with “9/45 (20%) were limited to the peripheral nervous system” because 4 cases affect PNS but also intracranial and spinal canal as it is stated in line 145;

Page 9

·         Lines 157,160 Standardize the indication of number of cases “…(6/22)…(5/24)” as used before in the text and not “(n=6/22)… (n=5/24)” “… (n=5/6)…. (n=3/5….)”;

  • It is suggested to change the legend of figures in general; they should indicate the location and pathological pattern of the lymphoma followed by the case number;

·         A suggestion for legend: “Figure 1. Intracranial lymphoma: intraparenchymal pattern. A) Case #11. Dog brain section……; B) Case #55. Feline brain section …….”

  • Line 184 Please indicate the designation of “HPF” before its abbreviation;

Page 10

  • Line 192 Standardize the indication of the number of cases (see comments for page 9 Lines 157-160);
  • Figure 2 A) Please use arrows pointing to the lesions. As suggested for Figure 1, change the legend accordingly. A suggestion of legend: “Figure 2. Intracranial lymphoma: Lymphomatosis cerebri. Case #39. A)……”;

Page 11

  • A suggestion for the legend of Figure 3: “ Figure 3. Intracranial lymphoma: A) Intravascular lymphoma. Case #30. Dog brain section with blood….. B) Meningeal lymphoma. Case #58. Cat brain…..”;
  • A suggestion of the legend of Figure 4: “Figure 4. Intracranial lymphomaleptomeningeal lymphomatosis. A) Case #40. Dog brain…. B) Case #49. Cat brain….”

Page 12

  • A suggestion for the legend of Figure 5: “Figure 5. Feline intraspinal lymphoma. A)…….”

Page 13

  • A suggestion for the legend of Figure 6: “Figure 6. Peripheral nervous system lymphoma. A)……..C) Case #9. Sciatic nerve of a dog…..”;
  • Line 323 Please indicate the 9 different pathological patterns of nervous system lymphoma identified to be clearer to the reader.

Page 14

  • Line 372 Regarding intravascular lymphoma (IVL) the authors did not discuss the results of the retrospective study with other references. The authors stated that the 10 IVL have been already reported in a previous study (reference 44), nevertheless, this pathological pattern is observed in almost half of the canine intracranial lymphomas, thus which should be minimally compared with other studies like for the other patterns. 
